# Germination of Microsporidian Spores: The Known and Unknown

**DOI:** 10.3390/jof9070774

**Published:** 2023-07-22

**Authors:** Qingyuan Huang, Jie Chen, Qing Lv, Mengxian Long, Guoqing Pan, Zeyang Zhou

**Affiliations:** 1State Key Laboratory of Resource Insects, Southwest University, Chongqing 400715, China; hqy550114547@email.swu.edu.cn (Q.H.); longmx@swu.edu.cn (M.L.); zyzhou@swu.edu.cn (Z.Z.); 2Chongqing Key Laboratory of Microsporidia Infection and Control, Southwest University, Chongqing 400715, China; 3Key Laboratory of Conservation and Utilization of Pollinator Insect of the upper reaches of the Yangtze River (Co-construction by Ministry and Province), Ministry of Agriculture and Rural Affairs, Chongqing Normal University, Chongqing 400047, China

**Keywords:** microsporidia, germination, spore wall proteins, polar tube, receptors

## Abstract

Microsporidia are a large group of mysterious obligate intracellular eukaryotic parasites. The microsporidian spore can survive in the absence of nutrients for years under harsh conditions and germinate within seconds under the stimulation of environmental changes like pH and ions. During germination, microsporidia experience an increase in intrasporal osmotic pressure, which leads to an influx of water into the spore, followed by swelling of the polaroplasts and posterior vacuole, which eventually fires the polar filament (PF). Infectious sporoplasm was transported through the extruded polar tube (PT) and delivered into the host cell. Despite much that has been learned about the germination of microsporidia, there are still several major questions that remain unanswered, including: (i) There is still a lack of knowledge about the signaling pathways involved in spore germination. (ii) The germination of spores is not well understood in terms of its specific energetics. (iii) Limited understanding of how spores germinate and how the nucleus and membranes are rearranged during germination. (iv) Only a few proteins in the invasion organelles have been identified; many more are likely undiscovered. This review summarizes the major resolved and unresolved issues concerning the process of microsporidian spore germination.

## 1. Infectious Parasites: Microsporidia

Microsporidia are a diverse group of fungal-related obligate intracellular parasites [1,2,3], whose life cycle consists of the proliferative phase (merogony), the sporogonic phase (sporogony), and the infective phase (mature spore) [4]. More than 1700 species and 220 genera of microsporidia have been described, which can infect invertebrates, vertebrates, and protists, while at least 17 species of microsporidia can cause infection in humans [5,6,7]. Microsporidia infections have been reported frequently in immunocompromised patients undergoing organ transplants or taking immunosuppressive medications [8]. Microsporidia have been identified as the cause of emerging and re-emerging infectious diseases by the US National Institutes of Allergy and Infectious Diseases (NIAID) and Centers for Disease Control and Prevention (CDC) as Category B biodefense priority pathogens [6]. Microsporidia also caused infectious diseases and serious economic losses in agriculture, aquaculture, and medicine. The microsporidia *Enterocytozoon hepatopenaei* infecting shrimp, *Nosema bombycis* infecting silkworm, as well as *Nosema apis* and *Nosema cerenae* infecting honeybee are four of the pathogens involved in Class I, II and III animal diseases issued by the Ministry of Agriculture and Rural Affairs of China in 2022 [9]. The transmission of microsporidia occurs in two different ways: horizontal transmission (primarily relies on the fecal-oral or urinary–oral route) and vertical transmission (direct transmission from parents to progeny) [10,11]. Animals infected by microsporidia display reduced body size, severe diarrhea, fewer offspring, and increased mortality [12,13]. 

Microsporidia possess fascinating parasite invasion mechanisms and regulation systems. The dormant, resistant mature spore (infective phase) has long been an area of considerable research interest. Due to their three layers of protective spore walls, microsporidia can survive without nutrients for years under harsh conditions [14]. Within seconds of attaching to host cells or receiving a stimulus, spores resume activity and begin to infect host cells in a unique process named germination [15,16,17]. Since the process of microsporidian invasion is unique, quite fast, and assumed to have little in common with the slow germination of fungal spores, “extrusion” was initially used to emphasize the rapid expulsion of sporoplasm from the microsporidian spores through the unique invasion apparatus, the polar tube (PT) [18]. Having gained a deeper understanding of the evolutionary relationships of microsporidia and fungi, in addition to the observation of some common characteristics, such as the activation and parts of germination processes, the term “germination” was frequently used [19,20,21].

Microsporidia primarily infect the host in the digestive tract and infect host cells through a unique invasion apparatus, the polar tube (PT) [22,23]. The dormant spores will reactivate metabolism, pierce or attach to the host cells with the fired PT, and deliver the infectious sporoplasm into the host cells [24,25,26]. The extreme resistance properties of spores will be lost once germinated, making them relatively susceptible to drug inhibition and further destruction [27,28,29]. Germination is a critical step in parasitism of microsporidia. Thus, there has been long-standing research interest in the spore germination, with researchers seeking a better understanding of this process to either find some efficient inhibitors to prevent spore germination or kill newly formed sporoplasm after germination. Although we have gained an increasing amount of insight into the molecular mechanism of microsporidian spore germination, there are still numerous unanswered questions.

## 2. Known about Microsporidian Spore Germination

The invasion apparatus of microsporidian spores is composed of the posterior vacuole, polaroplast, polar sac-anchoring disk complex, and the coiled polar filament (PF) [30,31]. PF changes to a hollow polar tube (PT) after being ejected from spores in a form similar to reversing the finger of a glove [32]. In addition, the protein-based exospore layer and the chitin-rich endospore layer play an important role in maintaining spore structure and tolerating swelling pressure during germination [31,33]. Inside the spore wall is the sporoplasm, which contains one or two nuclei and various cellular organelles. To complete the infection, the infectious cargo will be transported to the host cell through the PT after germination. Generally, spore germination occurs in four stages: (i) activation, (ii) an increase in intrasporal osmotic pressure, (iii) PF eversion, and (iv) sporoplasm passage through PT [34].

### 2.1. Activation of Spore Germination

Activators for spore germination differ greatly among different species of microsporidia [35,36,37]. In vitro germination of microsporidian spores has been extensively studied in the past few decades. Germination of spores is commonly activated by external pH changes, alkali metal ions, anions, and temperatures. Additionally, a significant difference was found between the germination conditions of two strains of microsporidia belonging to the same genus (*Trachipleistophora*) [38]. For many species of microsporidia, a shift in external pH to acidic or alkaline triggers germination [36,39,40,41,42]. When the spore is activated, some spore wall proteins (SWPs) can be shed, which may increase the permeability of the spore wall, reduce the rigidity of the spore apex, and lead to the ejection of the PT. For instance, SWP30 (SWP1) of *Nosema bombycis* was displayed both in the alkali-soluble proteins that were extracted from frozen spores and in proteins dissolved in 0.1 mol/L K_2_CO_3_ during germination, which was stimulated by the solution [43,44]. The protein was one of the main components of endospores, and its function is still unknown [43,45]. Although 36 SWPs (Table 1) have been identified so far, a large number of microsporidia SWPs have not been discovered [4,46]. Further studies of the diversity of these proteins can significantly improve our understanding of spore activation and invasion. Research on the proteins will enable us to deeper understand the mechanisms of spore wall assembly and the orderly arrangement of polar tubes, the molecular pathogenesis of microsporidian infection, and develop effective strategies to control microsporidiosis [14]. In addition, pH is studied in conjunction with other activators in order to gain a better understanding of how it affects germination. The spore wall functions as a barrier to prevent larger molecules from passing through, but alkali metal cations appear to pass freely through the spore wall and plasma membrane [47]. During germination, the smaller cations (e.g., Na^+^, and K^+^, which have been shown to trigger spore germination in *Nosema*) appear to be more efficient [20,36,47,48]. Spores of diverse microsporidia reacted differently to Na^+^, K^+^, and other cations during germination. The spore germination rate of *Antonospora locustae* (formerly *Nosema locustae*) has also been increased with LiCl, RbCl, and CsCl [36]. The germination of spores is also influenced by anions. The germination rate of spores was nearly 70% in 0.1 M NaCl or NaNO_3_, but reduced to 1% in Na_2_SO_4_ and Na_2_HPO_4_ [47]. Other anions have been used to promote PF discharge, including bromide, iodide, and fluoride [49]. The dehydration of some microsporidia followed by rehydration by hyperosmotic solutions has been found to be effective in promoting spore germination [36]. Low doses of ultraviolet radiation can trigger germination by disrupting the barriers between trehalose and trehalase [50]. Furthermore, microsporidian spores can germinate in response to changes in calcium ion concentrations, the osmotic pressure of the external medium, or in vivo host environments [42,51,52]. Intriguingly, studies have found that calcium chloride (0.001~0.1 mol/L) inhibits spore germination, whereas 0.2 M CaCl_2_ at pH 9.0 and 1 mM CaCl_2_ as well as calcium ionophore A23187 promote PF discharge [31,39,41,42,51,53,54,55,56,57,58]. It has been suggested that the displacement of calcium between the membrane and matrix of the polaroplast might be responsible for polaroplast swelling and the subsequent PF discharge [51,59]. According to this theory, the calcium ionophore A23187 triggers polaroplast swelling and discharge of PF, whereas calcium chloride inhibits it [59]. Calcium seems to be an essential component in regulating germination [31].

### 2.2. Energy of Germination

The first observation of the Microsporidia spore structure was made by Thélohan in 1894 [17]. In the years following the initial observations, studies have been conducted to define the internal structure of the spores. It is now generally accepted that in the phylum Microsporidia, spores can take a wide variety of shapes, from spherical to rodlike, and range in size from 1 to 40 μm [88,89]. In the anterior part of the spore, there is a system of membrane-limited cavities called polaroplasts. Typically, this structure occupies one-third to one-half of the spore volume, surrounds the straight part of the polar filament, and terminates at the level of the anterior polar filament coils. In mature spores, polaroplasts vary in shape but are usually lamellar [4,30]. And the posterior vacuole is a membrane-lined area filled with clear or spongy content [30]. The spatial relationship between polar filaments and posterior vacuoles is poorly understood. Different opinions exist regarding whether the filament enters or terminates at the vacuole. An image of the three-dimensional organization of the microsporidian shows the vacuole membrane interdigitating with the PF, cueing an interaction between these two organelles [16]. When microsporidia are exposed to activators, polaroplasts and posterior vacuoles appear to play an important role in the process of germination [20,48,52]. Hydrostatic pressure is now generally accepted as the reason for microsporidian spore germination. Pressure is due to an increase in water permeability or an increase in solute concentration [90]. Interestingly, the way osmotic pressure increases within species differs from species to species [91,92]. The decomposition of trehalose into glucose by trehalase in *Nosema algerae* can rapidly increase the intrasporal hydrostatic pressure inside the spore, which triggers spore germination [20,52]. Thus, aquatic microsporidia have been hypothesized to germinate through an increase in intrasporal osmotic pressure caused by trehalose degradation [93]. For terrestrial microsporidia, no changes in sugar content were observed after germination, so an alternative explanation is needed [93]. Studies indicated the presence of two crucial components of peroxisomal enzymes, catalase and acyl-CoA oxidase (ACOX), in the posterior vacuole of *Spraguea lophii* [94,95]. In the β-oxidation of the very long chain fatty acid (VLCFA) nervonic acid, the catalase and ACOX can convert H_2_O_2_ into water and oxygen [94,96]. As a result of the oxidation of long chain fatty acids and the subsequent production of molecular oxygen and water, the posterior vacuole may experience rapid swelling and cause germination [94]. Water inflow through aquaporins appears to be crucial to germination [97]. Recently, studies have shown that several aquaporins are located on the spore wall layer of *N. bombycis* and *Encephalitozoon cuniculi*. Moreover, the polyamine transporters or permeases encoded by some microsporidia may contribute to germination by absorbing sugars, cations, and nicotinic acids [98,99]. Altogether, increasing osmotic pressure induces spores to absorb water through functional aquaporins, resulting in swelling of the polaroplast and posterior vacuole [48,51,100]. The spore wall withstands osmotic pressure for some time but eventually ruptures at the apex, where it is thin, eventually leading to polar filament discharge [30].

### 2.3. Polar Tube and Eversion Process during Germination

When the microsporidia spore activates, accompanied by swelling of the polaroplast and posterior vacuole, the polar filament (PF) is fired from the mushroom-shaped anchoring disk (AD) and forms a hollow tube as an “everting finger of a glove” [32,48]. According to several studies, the everting PT is filled with electron-dense materials, which may be unpolymerized PTPs or tightly folded membranes [101,102,103,104]. Additionally, stretchability is a feature of the PT; the sporoplasm is transported through the PT into the host cell, and the diameter increases from 100 to 600 nm [16,32,34,104,105]. In spores, PF forms a right-handed helix, but the number of coils varies from species to species [16,18,106]. In *N. bombycis*, the structures of PF and PT have distinct structures; 881 and 1216 proteins have been identified from them, respectively [107]. Despite this, the structure and protein composition of the PF and PT remain a mystery. As of yet, according to research on the composition of the PT, six distinct PTPs (PTP1-PTP6) have been expressed and analyzed from different microsporidia (Table 2) [108]. PTP1, a proline-rich component of PT, has high tensile strength and elasticity. Consequently, it plays a crucial role in the discharge and passage of sporoplasm through narrow PT [109,110,111]. Moreover, as an O-linked mannosylated protein, PTP1 is capable of interacting with mannose-binding receptors on the cell membranes of its host [112,113,114,115]. According to further studies of PTPs, PTP2 is commonly found in various microsporidia at the same genomic locus as PTP1 and is more conserved than PTP1 [111]. PTP3, which can interact with other PTPs, might be a scaffold protein that contributes to the formation of PT [115,116]. PTP4 is found near the tip of the PT in *A. locustae* [6,108]. Subsequently, The PTP4 of *Encephalitozoon hellem* showed a similar location to the front of the PT, and immunoprecipitation analysis of PTP4 bound to host cell membranes identified transferrin receptor 1 (TfR1) as a host cell interacting partner for PTP4 [22]. In the genome, PTP4 and PTP5 are usually clustered, suggesting that they may have been linked in either evolution or expression [22,108]. A novel PT protein, NbPTP6, with cell-binding properties, was identified in *N. bombycis*. NbPTP6 was rich in histidine and serine, as well as having 6 O-glycosylation sites and 1 N-glycosylation site [117]. PTP6 homologs can be found in the genomes of other microsporidia [108]. In addition, ten potential novel PTPs in *N. bombycis* were screened [107]. Several NbPTPs were reported to interact with the spore wall proteins NbSWP5, NbSWP7, and NbSWP9 [67,69].

Many of the microsporidia proteins involved in spore firing are unknown. The anchoring disk is where the polar tube attaches to the spore wall, and it can rupture for PT discharge [118]. *E. cuniculi* spore wall protein EnP1 is embedded in the spore wall and abundant in the polar sac/anchoring disk region [61]. The homologous spore wall and anchoring disk complex protein NbSWP16 was identified in *N. bombycis* [74]. These proteins may play a role in the structural composition of the anchor disk and may also serve as substrates for proteases that promote the breakdown of the polar sac-anchor disk complex. NbSLP1, a subtilisin-like protease that localizes to both poles of the spore from *N. bombycis*, has been implicated in germination. NbSLP1 is active only at the apex of the spore, where the PT exits [119]. In further studies, intramolecular proteolysis has been shown to be necessary for NbSLP1 maturation, which undergoes a series of sequential cleavages at its N-terminus. The catalytic triad of NbSLP1 is crucial to its self-activation, as with *Bacillus amyloliquefaciens*, *Vibrio cholerae,* and other subtilisin-like enzymes [120,121,122]. Chitin is a major component of the endospore, which provides rigidity to the spore [4]. It has been demonstrated that the chitinase NbchiA from *N. bombycis* mainly hydrolyzes the second glycosidic linkage from the reducing end of (GlcNAc) 3–5, which may contribute to the discharge of polar filaments [123]. In teleost serum, chitinolytic and proteolytic activities may contribute to the defense against microsporidian infection [124].

**Table 2 jof-09-00774-t002:** The known proteins localized to the microsporidian polar tube.

Species	Protein/UniProtKB	Number of Amino Acids	Features	Mw(kDa)	References
*Encephalitozoon cuniculi*	EcPTP1/O76942	395	Acidic proline-rich, hydrophobicity, presence of tandem repeats, mannosylated with O-glycosylation, signal peptide, interact with the Concanavalin A (ConA), localized on the whole PT	37	Xu et al., 2004 [113] Bouzahzah et al., 2010 [114] Delbac et al., 1998 [125]
	EcPTP2/Q8SRT0	277	Basic lysine-rich core, acidic tail, can form intermolecular disulfide bridges with PTP1, RGD motif and signal peptide, localized on the whole PT	30	Delbac et al., 2001 [111] Peuvel et al., 2002 [116]
	EcPTP3/Q8MTP3	1256	Acidic core flanked by highly basic N- and C-termini, lacks cysteine residue, may assist in controlling PT extrusion, signal peptide, localized on the whole PT	150	Peuvel et al., 2002 [116]
*Encephalitozoon intestinalis*	EiPTP1/Q5F2J0	371	proline-rich, presence of tandem repeats, absent tryptophan and arginine, O-glycosylation and signal peptide, interact with the ConA, localized to polar filaments	35	Bouzahzah et al., 2010 [114] Peuvel et al., 2002 [116]
	EiPTP2/P0CAT4	275	lysine-rich, RGD motif, N-glycosylation and signal peptide	30	Peuvel et al., 2002 [116]
*Encephalitozoon hellem*	EhPTP1/O76273	453	proline-rich, presence of tandem repeats, mannosylated with N-, O-glycosylation, absent tryptophan and arginine, signal peptide, interact with the ConA, localized to polar filaments	43	Keohane et al., 1998 [110] Xu et al., 2004 [113] Peuvel et al., 2002 [116],
	EhPTP2/P0CAT5	272	lysine-rich, N-glycosylation, RGN motif and signal peptide, localized on the whole polar	30	Peuvel et al., 2002 [116]
	EhPTP4/I6UDI1	278	signal peptide, located at the tip of the PT, N-, O-glycosylation	36	Han et al., 2017 [22]
*Nosema bombycis*	NbPTP1/R0MQM8	409	signal peptide, O-glycosylation, phosphorylation, localized on the whole PT	55	Wu et al., 2014 [126]
	NbPTP2/R0KY97	277	PI = 9.39, signal peptide, interacted with SWP5, presence in the whole polor tube, phosphorylation, localized on the whole PT	39	Li et al., 2012 [67] Wang et al., 2007 [127] Yi et al., 2019 [128]
	NbPTP3/J7EQ15	1370	PI = 6.73, interacted with SWP5, localized on the whole PT	150	Li et al., 2012 [67] Wang et al., 2007 [127]
	NbPTP4/-	222	PI = 7.56, located at the front end of the PT and around the anchor disk, interact with Bmtubulin-α	25.3	Liu, 2019 [129]
	NbPTP5/R0KWI6	271	PI = 8.68, located at the front end of the PT and around the anchor disk	32.5	Liu, 2019 [129]
	NbPTP6/R0MBR8	247	rich in histidine (H) and serine (S), signal peptide, N-, O-glycosylation, cell-binding ability, localized on the whole PT	28.3	Lv et al., 2020 [117]
	NbSWP5/B3STN9	186	PI = 4.39, localized to the exospore and the region of the PT, interacts with the PT proteins NbPTP2 and NbPTP3	20.3	Li et al., 2012 [67,68]
	NbSWP7/B3STP1	287	PI = 4.78, located in the PT and spore wall.	32.8	Yang et al., 2015 [69]
	NbSWP9/R0MLT0	367	PI = 8.92, located in the spore wall, as well as anchoring disk and the front end of the PT after germination; interacts with NbPTP1 and NbPTP2	42.8	Yang et al., 2015 [69]
*Enterocytozoon hepatopenaei*	EhpPTP2/A0A1W0E7X7	284	PI = 8.8, rich in lysine, super family domain (pfam17022), O-glycosylation	32	Wang et al., 2021 [130]
*Antonospora locustae* (formerly *Nosema locustae*)	AlPTP1/-	355	PI = 5.2, rich in proline and glycine, N-, O-glycosylation, interact with the ConA, localized on the whole PT	34	Polonais et al., 2005 [131]
	AlPTP2/C8CG41	287	PI = 9.1, signal peptide, O-glycosylation, rich in lysine, localized on the whole PT	29	Polonais et al., 2005 [131],
	AlPTP2b/C8CG42	568	signal peptide, PI = 8.4, O-glycosylation, rich glycine and serine, b-turn structures, localized on the whole PT	55	Polonais et al., 2013 [132]
	AlPTP2c/C8CG43	599	signal peptide, PI = 8.7, O-glycosylation, rich glycine and serine, b-turn structures	56	Polonais et al., 2013 [132]
*Paranosema grylli*	PgPTP1/-	351	PI = 5.2, acidic and proline-rich, N-glycosylation, interact with the ConA	33	Polonais et al., 2005 [131]
	PgPTP2/-	287	PI = 8.9, rich in lysine	29	Polonais et al., 2005 [131]
*Nosema pernyi*	NpPTP1/A0A482G4U9	394	PI = 5.82, signal peptide	39.16	Wang et al., 2019 [133]
	NpPTP2/A0A482G3T3	277	PI = 9.39, signal peptide	30.8	Wang et al., 2019 [133]
	NpPTP3/A0A0N7ABT9	1370	PI = 6.52, localized on the whole polar	148.56	Wang et al., 2019 [133]

The whole process of PF eversion in *E. hellem* and *E. intestinalis* takes less than 500 ms, while that of *Anncaliia algerae* takes 1.6 s [16]. Generally, the process of PF eversion can be divided into three phases: (i) PT elongation, (ii) a static phase, where the PT length does not change; and (iii) the emergence of cargo at the distal end of the PT [16]. In the initial phase of PF eversion, PT emerges from the apical portion of the spore and elongates to its maximum size [16,48]. PT length and width may differ among species, but the fully extended PT is much longer than the PF that is inside the spore, suggesting that either the PT is stretched or that the protein or stacked membrane subunits that make up the PT undergo a reorganization during eversion, and once the PT has reached 50% extension, sporoplasm transport begins [16,104]. The stationary phase begins after the PT reaches its maximum length. Microsporidia are mainly transporting sporoplasm at this stage, and the rate of transport varies significantly among different species. During this process, the spore nucleus undergoes significant deformation when passing through the tube due to the significant difference in diameter between the nucleus and the PT. Interestingly, a pause in nuclei movement was found approximately three-quarters of the way through the PT [16]. Currently, there are two hypotheses: (i) During extension, the end of the PT may remain closed, and the delay in the opening of the distal end of the PT may be responsible for the observed pausing during nuclear translocation [48]. (ii) PT firing and sporoplasm transport may diminish the driving force, followed by a further increase in force to complete sporoplasm export, suggesting that sporoplasm may be pushed out of the tube by a new force [108]. In phase 3, the sporoplasm is extruded from the PT at the distal end and has an approximate circular shape. The PT components may interact with the sporoplasm in a specific way, or some membrane from the sporoplasm may remain inside the tube to form an adhesion bridge with the PT [16].

## 3. Spore Germination: Major Unanswered Questions

Despite much progress in understanding spore germination in recent years, there are still many fundamental questions. Answers to these questions will be crucial to a detailed understanding of spore germination. We will discuss these major questions in the following sections.

### 3.1. What Is the Initial Germination Signal and Receptor of Microsporidia?

In general, the germination of microsporidian spores must be modulated by specific conditions so as to prevent premature and unproductive germination outside the target cells [52,118]. Additionally, conditions that activate spores vary widely among species, presumably reflecting the organism’s adaptation to their host and external environment [35,36]. Considering the speed and irreversibility of germination, it may be the result of irreversible enzyme metabolism since such changes accompany swelling of the posterior vacuole. Although circumstances in vivo are still unclear, a variety of conditions have been studied in vitro to promote germination [31,36]. pH changes, alkali metal ions, and other treatments can markedly increase spore germination ratios. However, it is still unclear which proteins are initially activated by these activators. Since microsporidia are activated by external signals, membrane proteins may play a crucial role in receiving environmental stimuli, and some components of the spore wall need further study. The mechanosensitive ion channel (MscS) is one of the membrane proteins that controls intracellular pressure by responding to extracellular signals in both prokaryotes and eukaryotes [134]. Thus, during microsporidian germination, microsporidian-encoded MscS may be able to receive external signals and regulate the signal transduction system [135]. According to Dall [90], the alkaline external medium creates a proton gradient, which regulates the flow of water in the spore and increases intrasporal pressure by regulating proton-cation exchange through the cation transporter to stimulate spore germination. Additionally, PT firing seems also to be affected by Ca^2+^ that crosses the spore wall through ion channels and interacts with receptors on the spore wall or the cytoskeletal components [42,136]. Germination may also be mediated by other membrane proteins, such as Major Facilitator Superfamily (MFS) transporters [137] and nucleotide transporters, NTT proteins [138]. In addition, it has been shown that the glycosaminoglycans (GAGs) on the host cell surface may act as a non-specific receptor to stimulate microsporidia germination in vivo [139]. Further studies show that the attachment of microsporidian spores to GAGs is facilitated by exogenous divalent cations [140]. However, the specific proteins that receive signals from external media have not been identified.

### 3.2. How Do the Germination Signals Stimulate Microsporidian Polar Filament to Eject?

Transcriptomic and proteomic analyses have been performed in recent years to determine how microsporidia germination signals induce PT ejection [25,26,141]. A total of 541 significantly differentially expressed genes (DEGs) were identified through differentially expressed gene analysis of mature spores and sporoplasms in *N. bombycis*. A total of 29 KEGG pathways were enriched among the DEGs, including the ribosome and pentose phosphate pathways, carbon metabolism, aminoacyl-tRNA biosynthesis, amino sugar and nucleotide sugar metabolism, ubiquitin-mediated proteolysis, and purine metabolism, preparing the host for invasion and proliferation by generating energy and synthesizing substances [141]. In another study, RNA-Seq and immunoblot analyses identified a protein phosphatase-associated gene as significantly up-regulated, which suggests that microsporidia germination may be impacted by dephosphorylation. Liu et al. [25] speculated that some proteins involved in spore germination may be dephosphorylated by the protein phosphatase PP2-A and lose their function of maintaining dormancy. Based on high-resolution mass spectrometry (MS) data in *N. bombycis*, 127 proteins were significantly different among germinated and non-germinated spores. In particular, PTP1 and PTP3, SWP4, and SWP30 were found to be significantly down-regulated, and SWP9 was significantly up-regulated. Additionally, polynucleotide kinase/phosphatase and flap endonucleases 1, as well as a panel of hydrolases related to protein degradation and RNA cleavage, were highly expressed too, which indicates that these enzymes play a role in spore germination [26]. The transcriptome analysis of non-germinated and germinated spores of *E. hepatopenaei* identified 137 of 2536 functional genes with significant differences in terms of cellular metabolism, transcription, and other aspects. Based on a KEGG annotation analysis, 28 pathways were identified as being involved in the process from ungermination to germination, primarily Translation, Cell growth and death, Folding, Sorting, Degradation, and Replication and Repair [142]. However, spore germination is an intricate and elaborate process. With current information, it is difficult to elucidate how signaling occurs for a few seconds.

### 3.3. How Do Polar Tube Ejections Happen and Where Does the Energy Come from?

Currently, it is unclear what causes and drives the ejection of PT from spores. Several explanations have been proposed over the years as possible explanations for this rapidly occurring and mysterious process: (i) The reversing finger of a glove is an analogy for PT firing that involves the PT turning inside out as it is extended [18,32]. The PT reverts after the anchoring disk (AD) ruptures, while the collar-like structure in the AD turns and holds the tube in place during the rupture [35]. In additional research, it has been shown that the membrane of the extrusion apparatus exteriorizes and becomes the outer envelope of the fired sporoplasm [94]. By fully everting the extrusion apparatus, the posterior vacuole contents are discharged into the external medium, where the sporoplasm is contained by the everted membrane of the polaroplast. On the spore wall, the original membrane remains. It is believed that this process is powered by the posterior vacuole, which expands dramatically when the sporoplasm is released [94]. (ii) The discharge of the PT can be compared to the extension of a telescope. According to the model, the PT is completely constructed before eversion, and the extension of the extruded PT results from the PT unfolding [143]. An examination of the extruded PT of *A. algerae* using cryo-transmission electron microscopy (CTEM) revealed that the PT contains masses of tightly folded or stacked membranes [101], which may be unfolded during the ejection of the PT. (iii) Another theory suggests that PTPs are stored at the growing tip to lengthen the PT. This hypothesis suggests that unpolymerized PTPs are deposited at the growing tip of the filament during eversion from the dense core [48,59,144,145]. (iv) PT is attached to or continues the sporoplasm. In the process of germination, as with a compressed spring, the PT could be released from the spore [146,147]. The firing model of “jack-in-the-box” allows the PT to be extremely distorted and stressed within the spore. As the PT is pushed out of the spore, its coils around itself are released, and it becomes a linear tube [147]. Despite the fact that each of these models explains how PT ejection occurs, it should be noted that each hypothesis has flaws. The first and second hypotheses suggest that the swelling of the posterior vacuole is the main driving force behind the PT ejection. Video of the PT firing of *Edhazardia aedis* shows the posterior vacuole swelling primarily after the ejection of the PT, which is thought to be involved in cytoplasm expulsion [148]. In this way, it is possible for PT ejection to be driven by other energies. Flexibility is a feature of the PT, but further experimental evidence is lacking for the spring model. There is still a lack of clarity regarding the process based on current hypotheses and evidence.

### 3.4. How Do Nuclei Pass through the PT?

Nuclei vary in size during the life cycle of microsporidia, and in mature spores, the nucleus is usually 1–2 μm or smaller [30,149]. Incredibly, the diameter of microsporidian PT like *Ameson michaelis* is only 100–120 nm [104]. Thus, when the nucleus travels through the PT, it undergoes extraordinarily fast and large deformations. Stresses may affect microsporidian chromatin due to the extent of nuclear deformation during transport [16]. The composition of nuclear laminae and the organization of chromatin are associated with deformability [150]. Lamin expression controls nuclear stiffness and deformability in metazoan cells, e.g., hematopoietic, immune, and cancer cells [151,152,153]. It has been reported that losing lamin in metazoans increases nuclear fragility and deformability [154]. The sequence of the microsporidian genome has not yet revealed any lamins or homologs [16,155,156], which might explain why nuclear deformation occurs to such a high degree and so rapidly in microsporidia. Breast cancer cells should undergo a series of deformations during migration. Due to the loss of a nuclear lamina component, breast cancer cells have been found to have a breakdown of their nuclear envelopes while migrating. Notably, the timescale for microsporidian nuclear transport is much faster than the migration of mammalian cells [152,157]. Further studies are needed to examine how the nuclear state is regulated during firing and how the nuclear membrane is protected.

## 4. Conclusions

A fascinating aspect of microsporidia is their peculiar way of infecting host cells, which begins with spore germination, and this process may be regulated by receptor proteins on cell membranes and external signals. In a few hundred milliseconds following the swelling of the posterior vacuole, PT undergoes a significant conformational change, allowing the passage of infectious sporoplasm into a host cell. The activators responsible for spore germination initiation differ greatly among microsporidia species, and despite intensive research into the spore germination conditions of some microsporidia, no comprehensive research has been published, so our understanding of this process is limited. Furthermore, these in vitro conditions can mimic some physiological conditions in the host, but it remains unclear how microsporidia germinate once they enter the host. After PT germination begins in the hundreds of milliseconds, there is no comprehensive understanding of how other organelles, particularly the nucleus, deform and move through the PT inside the spore. Research in the past hundreds of years has provided a wealth of new information about microsporidia germination since the first observation in 1894 using light microscopy [17]. Researchers have developed many methodologies to analyze the germination of single species of spores among populations and have compiled vast amounts of bioinformatics data in transcriptional, proteomic, and genome sequence studies. Still, major, fundamental questions remain regarding the function, structure, and regulation of proteins and activators important for spore germination in vivo (Table 3 and Figure 1). Cracking the microsporidia germination code will be useful in the development of a treatment for microsporidiosis.

## Figures and Tables

**Figure 1 jof-09-00774-f001:**
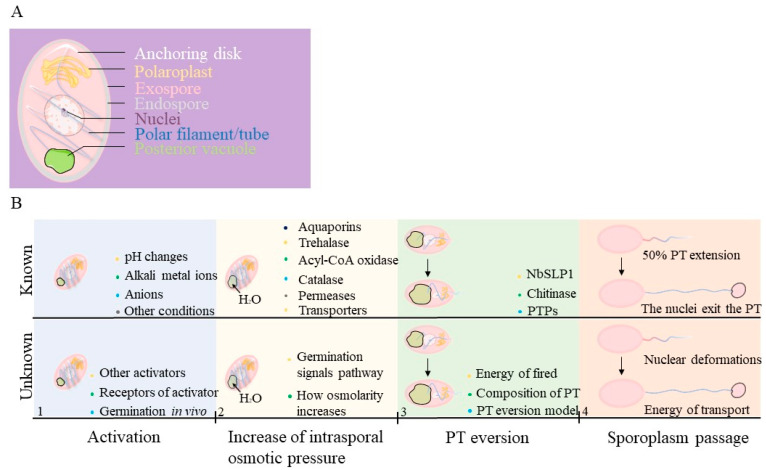
PT germination and sporoplasm transport model. (**A**) The schematic diagram shows the spatial organization of organelles involved in microsporidian spore germination. Polaroplasts (yellow) surround the anterior part of the PT (blue), which is joined to the anchoring disk (white). The nucleus (purple) and the posterior vacuole (green) are surrounded by the right-handed helix of PT. (**B**) Known and Unknown information during the process of microsporidian spore germination. See Table 3 for a detailed explanation.

**Table 1 jof-09-00774-t001:** The identified microsporidian spore wall proteins.

Species	Protein/UniProtKB	Number of Amino Acids	PI	Mainly Location	Features	Mw (kDa)	References
*Encephalitozoon cuniculi*	EcSWP1/Q9XZV1	450	4.96	Exospore	glycine- and serine-rich repeats	51	Bohne et al., 2000 [60]
	EcEnP1/Q8SWL3	357	9.07	Endospore	HBM	40.5	Southern et al., 2007 [61] Peuvel et al., 2006 [62]
	EcEnP2/EcSWP3/Q8SWI4	221	8.42	Endospore	glycosylphosphatidylinositol (GPI) anchored and O-glycosylation sites	22	Peuvel et al., 2007 [62] Xu et al., 2006 [63]
	EcCDA/Q8SU65	254	4.43	Endospore,plasma membrane	Glycoside hydrolase and deacetylase	28.1	Brosson et al., 2005 [64]
*Encephalitozoon intestinalis*	EiSWP1/Q95WA3	388	4.78	Exospore	-	41	Hayman et al., 2001 [65]
	EiSWP2/Q95WA4	1002	3.68	Exospore	Repeating of amino-acid units	107	Hayman et al., 2001 [65]
	EiEnP1/A7TZU4	348	8.84	Exospore,endospore and polar membrane layer	HBM	39.1	Southern et al., 2007 [61]
*Encephalitozoon hellem*	EhSWP1a/C3VJR1	509	4.30	Exospore	-	55	Polonais et al., 2010 [66]
	EhSWP1b/C3VJR2	533	4.64	Exospore	-	60	Polonais et al., 2010 [66]
*Nosema bombycis*	NbSWP1/B3STN5	278	7.95	Endospore	-	30.4	Wu et al., 2008 [45]
	NbSWP2/B3STN6	268	8.45	Endospore	HBM	25.3	Wu et al., 2008 [45]
	NbSWP3/B3STN7	316	7.29	Exospore	-	32.7	Wu et al., 2008 [45]
	NbSWP5/G0Z414	186	4.39	Endospore and PT	Interacts with PTP2 and PTP3	20.3	Li et al., 2012 [67,68]
	NbSWP7/B3STP1	287	4.78	Exospore and endospore	Interacts with NbSWP9	32.8	Yang et al., 2015 [69]
	NbSWP9/R0MLT0	367	8.32	Exospore, endospore and PT	Interacts with PTP1, PTP2 and NbSWP9	42.8	Yang et al., 2015 [69]
	NbSWP11/B3STP5	446	9.27	Exospore and endospore	DnaJ domain and HBM	52.3	Yang et al., 2014 [70]
	NbSWP12/B3STP6	228	6.78	Exospore, endospore, membrane of meront	BAR-2 domain and HBM	26.6	Chen et al., 2013 [71,72,73]
	NbSWP16/R0MN98	211	8.42	Exospore	HBM and proline-rich tandem repeats	44	Wang et al., 2015 [74]
	NbSWP26/B9UJ97	223	5.09	Exospore	HBM and N-glycosylation sites	25.7	Li et al., 2009 [75]
	Unnamed/EOB13250	244	10.24	Endospore	Transmembrane domain	28	Wang et al., 2020 [76]
*Nosema ceranae*	Unnamed/A0A0F9WE74	226	6.84	Endospore	-	26.19	Liang et al., 2021 [77]
	NcSWP8/A0A0F9WIV3	172	4.00	-	Promote proliferation	19.5	He et al., 2021 [78]
	NcSWP12/A0A0F9WTX8	229	7.88	-	Promote proliferation	26.7	He et al., 2021 [78]
*Enterocytozoon hepatopenaei*	EhSWP1/A0A1W0E3P7	228	8.45	Exospore and endospore	Exospore and endospore	27	Jaroenlak et al., 2018 [79]
	EhSWP2/A0A1W0E3S3	228	5.12	-	MICSWaP domains	25.7	Li et al., 2021 [80]
	EhSWP3/A0A1W0E914	249		Exospore and endospore	transmembrane domains	27.1	Fan et al., 2022 [81]
	EhSWP7/A0A1W0E705	250	5.04	-	-	25.3	Li et al., 2021 [80]
	EhEnp1/A0A1W0E696	333	8.86	-	-	38.3	Li et al., 2021 [80]
*Antonospora locustae* (formerly *Nosema locustae*)	AlocSWP2/A0A1W0E3S3	222	5.16	Exospore and endospore	HBM	25	Chen et al., 2017 [82]
	EbSWP1/B7XHM5	228	7.06	-	O-linked glycosylation site	26.8	Meng et al., 2022 [83]
*Enterocytozoon bieneusi*	EbSWP2/B7XJH4	247	9.46	-	N-linked glycosylation sites	29.2	Meng et al., 2022 [83]
	EbSWP3/B7XHL8	229	5.15	-	N-linked glycosylation sites	25.9	Meng et al., 2022 [83]
	NpSWP1/A0A060A4C2	278	7.02	Endospore	Transmembrane	32	Zhu et al., 2014 [84]
*Nosema pernyi*	NpSWP9/A0A0N7AC01	317	5.75	-	-	37.16	Ma et al., 2017 [85]
	NpSWP12/A0A0S2EGT8	228	5.96	-	BAR domain	26.6	Feng et al., 2015 [86]
*Nosema antheraeae*	NaSWP8/G3CU65	161	4.802	-	HBM	18.4	Xi et al., 2010 [87]

**Table 3 jof-09-00774-t003:** Microsporidian spore germination—known and unknown information.

Germination Stages	Known Information	Information Gaps
Activation	Spores may be triggered by pH changes, alkali metal ions (e. g. Na^+^, K^+^, Ca^2+^), anions (e. g. Cl^−^, CO_3_^2−^, Br^−^), and other stimuli	The mechanisms by which receptors are triggered in vivo and the path of signaling in activation.
Increase of intrasporal osmotic pressure	An increased intrasporal osmotic pressure may be caused by trehalase, acyl-CoA oxidase, catalase, permeases, and some transporters. As soon as the spores absorb water through functional aquaporins, swelling of the polaroplasts and posterior vacuoles occurs.	The main substances that cause changes of intrasporal osmotic pressure. A challenge in understanding osmolarity increases is identifying the signaling pathway.
PT eversion	The subtilisin-like protease or chitinase may play a key role during the PT firing occurs.	The details of the PT structure, the energy source, and the ejection of the PT are still unclear.
Sporoplasm passage	As soon as the PT extend a half, the nuclei deform to fit into the PT and exit from the spore coat. Vesicles can be observed in the extruded PT. Sporoplasm returns to a circular shape in the tip of PT.	The sources of energy, regulation of the nuclear phases and deformation of sporoplasm are difficult to understand.

## Data Availability

Not applicable.

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
