# Peer review of "Germination of Microsporidian Spores: The Known and Unknown"

_jof, 2023, doi:10.3390/jof9070774_

Round 1

Reviewer 1 Report

This paper clearly shows the germination of the microsporidia and the point of unraveling so far. I judge that this paper is worth publishing this paper in JoF. However, there is a statement that does not follow the flow of some text, and there is a Figure and Table that is difficult to understand. I think that improving these will be a better paper. Please consider the corrections below.

L.315-335)'' 3.4 How Do Nuclei Pass Through The Pt '' significant modification

Here, a story of fungi, which have infectious abilities in plants, comes out. This fungus is similar to Microsporidia, so the authors show it as an example, but are very abruptly described in terms of the flow in the text. It is necessary to clearly show the relationship with MicroSpiridia and how it is a little more similar. It seems there is no problem if you delete the description as long as you read it.

Fig.1

This figure is the most important part of this paper. However, the expression of Fig.1 does not show it to understand it properly. Create a new Table and explain it clearly. Also, if you add the illustration of Fig.1 to the side of the table, it will be easier to see.

Reviewer 2 Report

The ms of Huang with co-authors is an excellent review devoted to the very actual topic of the modern microsporidiology. It is a successful summation of available information about the organization of the invasion apparatus of microsporidia with emphasis on the proposed mechanisms of germination of microsporidian spores. The term ‘germination’ was earlier infrequently used among microsporidiologists since it was assumed that the process of microsporidian invasion is unique, very fast and has little in common with the slow germination of fungal spores. The term ‘extrusion’ was coined and used instead. Recently, as our understanding of the joint evolutionary history of microsporidia and fungi deepened, the term ‘germination’ began to be used more often. In the ms of Huang et al. the specificity of terminology is not discussed. The still widely used term ‘extrusion’ is almost not mentioned in the review, only in section 3.3 and in the bibliography. I would advise the authors to discuss these terminological nuances at the beginning of the review.

Also, the authors need to explain more clearly the difference in use the terms ‘polar filament’ (before discharge) and ‘polar tube’ (after discharge), and when the first one turns into the second one. It is necessary to ensure that these terms are used consistently in the text and are not mixed and confused (e.g. check at the beginning of paragraph 2).

            When the authors describe the organisation of the invasion apparatus, they do not mention an important component - a polar sac-anchoring disk complex:

“The invasion apparatus of microsporidian spore is composed of the posterior vacuole, polaroplast, and the polar filament or PT [27].” The polar sac-anchoring disk complex should be also included in the list of organelles of invasion apparatus (see Vávra, J.; Ronny Larsson, J.I. Structure of Microsporidia. In Microsporidia; Weiss, L.M., Becnel, J.J., Eds.; John Wiley & Sons, Inc.: Chichester, UK, 2014; pp. 1–70, ISBN 978-1-118-39526-4).

Below I provide the list of my remarks and concerns (including small editing remarks to the text:):

17: extrudED PT

31-32: 220 species genera

39: immunoreacted patients – Does it mean ‘immunocompromised patients’?

45: or cytokinesis into them – this phrase is unclear

48: vertical transmission (direct transmission from parents to progeny)

53-54: making them relatively susceptible to drug-inhibited and further to be destroythis phrase is unclear, consider revision, e.g. ‘making them relatively susceptible to drug-inhibition and further destruction’

69 (missing gaps): (ii)_an increase…., (iii)_PT eversion

76-81 (the following editing is suggested): When spore is activated, some spore wall proteins (SWPs) can be shed, it may increase permeability of the spore wall, reduce the rigidity of the spore apex, and lead to the ejection of the PT. For instance, SWP30 (SWP1) of Nosema  bombycis were displayed both in the alkali-soluble proteins which were extracted from frozen spores and in the germinated supernatant simulated by potassium carbonate [37,38]. The protein was one of the main components of endospores, and its function is still don’t unknow  the function [37,39].

the germinated supernatant simulated by potassium carbonate this phrase is not quite clear, please, add some explanatory details.

82-83: Although 36 SWPs (Table 1) have been identified so far, a large number of microsporidian SWPs have not been discovered which can significantly improve our understanding of spore activation and invasion [4,40]consider revision (e.g. further studies of the diversity of these proteins can significantly improve…)

88-89: During germination, the smaller cations appear to be more efficient, such as e.g. Na+ and K+ which have been shown to trigger spore germination in Nosema [31,41-43].

103-104: It has been suggested that the displacement of calcium (what does it mean?may be ‘release of calcium’?) from the polaroplast membrane (may be ‘from the polaroplast compartment’?) might either activate a contractile mechanism or combine with the polaroplast matrix (what does it mean?) to cause spore (what does it mean? may be 'sporoplasm'?) release [46,54].

141-143: AnoOther studies found that the posterior vacuole of Spraguea lophii spores contained catalase and acyl-CoA oxidase (ACOX), which are crucial component of peroxisomal enzymes

144: the catalase and ACOX can convert H2O2 breakdown into H2O and O2

161: flexibility? – may be more suitable term here is stretchability (or tensility)?

168-169 (gaps): research on the composition of the PT, six distinct PTPs_(PTP1-PTP6) have been expressed and analyzed from different microsporidia_(Table 2) [108].

193-194: The catalytic triad of NbSLP1 is crucial to its self-activation, as with in case (?) other subtilisin-like enzymes [119].

283: It remains unclear what mechanism drives and energy source PT ejection from the spores. – consider revising of English

285: enigmatically process

294: ‘when spores are released discharged’  or ‘when sporoplasmes are released’ ?

306-307: Despite the fact that each of these models explains how the PT ejections occur. – unfinished sentence, consider revising. Probably it should be combined with the following one.

317: Incredibly, Tthe diameter…

341: allowing the passage of Iinfectious polaroplast? into a host cell –probably you mean ‘the passage of infectious sporoplasm’?

361 (missing gap): Nucleus_(purple)

Figure 1 (a). light blue signature on a lilac background is hard to see, it is better to choose a different color. In the labeling “Polar tube/Filaments” – filament should be in singular: “Polar tube/filament”

Information about doi in the References is missing. The following references should be formatted: 41, 45, 47. 48, 50, 52, 87, 93. Please, indicate English title of the journal in reference 48.

Minor editing of English is desirable (see my comments in the previous box).

Reviewer 3 Report

The paper is a narrative review concerning the process of germination of microsporidia. The topic is of interest and the manuscript can be accepted after a major revision, concerning the English form, also.

please cite and comment the following references DOI: 10.3390/insects14020185 , 10.14411/fp.2020.023, 

lines 31-32 -  the sentence "1700 and 220 species" is not correct, please, modify 

line 45 - please, verify the sentence

line 54 - to be destroyed

line 57 - please, change "found" in "find"

line 64 - role

line 72 - among

line 102 - promotes

Table 1 Antonospora locustae (line 177, also) is a synonym of Nosema locustae (see line 91). Please, add information

line 120 - Microsporidia

line 195 - please rewrite in a more proper mode. In detail change microsporidia in microsporidium (is not a plural) and make understandable the relationship between turbot with Nosema bombycis

line 317 - the

line 327 - please modify the statement. Mammalian are metazoa

line 149 - Encephalitozoon cuniculi, please, at the first mention genus name should be reported per extenso.

line 333 - please, delete full stop

line 336 - may be "Conclusions" instead of "Summary" would be more proper

line 341 - infectious

line 347 - "milliseconds" is written elsewhere as "ms", please adjust

The English form should be revised, mostly in the first pages. Some suggestions have been reported (see comments and suggestions for Authors)

Round 2

Reviewer 1 Report

This manuscript has been properly revised.

Reviewer 3 Report

The manuscript has been revised according to reviewer's suggestions